# Least Squares Reverse Time Migration of Ground Penetrating Radar Data Based on Modified Total Variation

**Qianwei Dai** [1,2], **Shaoqing Wang** [1,2] **and Yi Lei** [3,*]

1 Key Laboratory of Metallogenic Prediction of Nonferrous Metals and Geological Environment Monitoring, Changsha 410083, China; qwdai@csu.edu.cn (Q.D.); csuwsq@csu.edn.cn (S.W.)
2 School of Geosciences and Info-Physics, Central South University, Changsha 410083, China
3 School of Civil Engineering, Central South University, Changsha 410075, China
* Correspondence: leiyi862357@csu.edu.cn

**Abstract:** As a fundamental part of ground penetrating radar (GPR) data processing, reverse time migration (RTM) can correctly position reflection waves and focusing diffraction waves on the proper spatial position. Least-squares reverse-time migration (LSRTM) is widely used in the seismic field for its ability to suppress artifacts and generate high-resolution images in comparison to conventional RTM. However, in the particular case of GPR detection methods, LSRTM is extremely susceptible to aliasing artifacts caused by under-sampling. In pursuit of enhanced precision in underground structure characterization, this paper presents the development of a new LSRTM based on modified total variation (MTV) regularization to improve imaging resolution. Initially, the objective function of LSRTM is derived by combining the Born approximation in 2-D transversal magnetic mode. Next, the adjoint equations and their gradients are solved using the Lagrange multiplier method. The objective function is then constrained by MTV regularization to ensure the precision and convergence of the LSRTM, which delivers a refined edge with reconstruction details. In the numerical experiments, in comparison to the conventional LSRTM method, the LSRTM-MTV algorithm demonstrated a 30.4% increase in computational speed and a 21.1% reduction in mean squared error (MSE). The outperformance of the proposed method is verified in detail through the image resolution and amplitude preservation in the test of synthetic data and laboratory data. Future research efforts will center on applying the proposed method to models featuring dispersive or anisotropic media that closely mimic real-world conditions and extending the application to various imaging techniques involving objective function minimization.

**Keywords:** ground penetrating radar; least-squares reverse-time migration; modified total-variation regularization

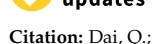



## 1. Introduction

Ground Penetrating Radar (GPR) is a fast, high-resolution, nondestructive detection technology which has been widely used in various fields such as building structure investigation [1,2], environmental research [3], engineering detection [4], etc. Due to the complex and varied distribution of the underground medium, multiple waves, including diffraction waves, are usually produced when electromagnetic waves encounter a contrast of electrical properties. In most cases, the GPR measurement area is filled with various electromagnetic interferences; the noise brought on by the surrounding environment will contaminate the collected GPR observed signal, such that the GPR profile is unable to accurately reflect the underground structure, thus requiring methods to improve the imaging solution [5]. Migration typically provides an efficient method for imaging complicated structures. It can properly position the reflection waves and focus the diffraction waves backward to their proper spatial position [6,7], and the distribution of underground structures can be obtained effectively [8] using finite-difference migration [9], Kirchhoff integration migration [10],

F-K migration [11], reverse time migration (RTM) [12–14], etc. The RTM method, based on two-way wave equations, properly utilizes full-wavefield information and possesses the unique advantages of high precision, accurate phase, and propagation in any specific direction [15,16], which is suitable for imaging complex structures [17]. However, the RTM method remains only the conjugate transpose of the forward operator, not its inverse, and the result of the RTM method does not represent the reflection coefficient. For example, RTM images often contain various undesirable artifacts when sensor bandwidth and coverage are limited, which can affect the imaging resolution of underground structures including possible anomalies. Furthermore, conventional RTM results usually contain only information on the location of underground structures without providing sufficiently precise amplitude information, which makes it impossible to obtain an accurate velocity distribution of the subsurface medium [18].

Least-squares reverse-time migration (LSRTM) solves RTM imaging as a least-squares inversion problem and achieves an imaging result that best matches GPR observation data by implementing an iterative algorithm [19]. Based on the observational data space, the method is generally employed on ray class and wave equation continuation operators to perform high-resolution subsurface imaging. However, in cases of severely degraded data quality, LSRTM imaging results contain artifacts that usually cause inversion instability and slow convergence of the inversion [20]. Since LSRTM involves solving the linear inverse problem, several different approaches have been developed to improve LSRTM imaging. For instance, quadratic and non-quadratic regularization (Cauchy norm) have been incorporated into LSRTM to improve resolution while suppressing spatial artifacts [21]. Qp-LSRTM has been employed to rectify distortion caused by strong subsurface attenuation. The merit of this approach compared with conventional RTM and LSRTM is that Qp-LSRTM compensates for the amplitude loss due to attenuation and produces images with better balanced amplitudes and more resolution below highly attenuative layers [22]. Furthermore, Tikhonov regularization is employed alongside LSRTM for ensuring the stability of the inversion. However, Tikhonov regularization usually tends to over-smooth the boundary between the target region and the background region [23]. Instead, approximate Total Variation (TV) regularization based on image reconstruction helps to recover the discontinuous image efficiently and has many other advantages. For example: limiting the smoothness of the image, eliminating the stair effect, avoiding edge blurring, etc. [24]. However, the TV regularization constraint method is exceptionally sensitive to smoothing variables due to the convergence of the inversion as well as the nonlinearity caused by the data fitting term and the total variation regularization term [25].

To address noise suppression, amplitude preservation, and resolution improvement, this paper presents the development of an LSRTM approach based on modified total variation (MTV) regularization [26]. The objective function is then constrained by MTV regularization to ensure the precision and convergence of the LSRTM, which delivers a refined edge with reconstruction details. Finally, the proposed method is assessed with theoretical synthetic data and laboratory data to confirm the efficacy and applicability of our algorithm.

## 2. Materials and Methods

### 2.1. LSRTM

Compared to Kirchhoff migration and one-way wave equation migration, RTM offers the benefits of high imaging accuracy and no limitation of interface inclination [27]. However, there are also deficiencies, such as high dependence on the velocity model, low imaging resolution, and high data requirements. As an improvement, LSRTM updates the coefficient of reflection iteratively through the idea of inversion, which has a higher resolution image than RTM. This can be attributed to the inversion feature of LSRTM. Compared to other migrations, LSRTM is essentially a waveform inversion of the output reflection coefficient with the advantage of linear inversion.

Here, we briefly introduce the workflow of LSRTM for electromagnetic wave equations. Firstly, the reverse migration operator is derived based on the Born approximation. The given background electrical medium and the background electrical field of the waves in 2-D TM$_z$ mode are satisfied [28]:

$$\frac{\partial H_x}{\partial t} = -\frac{1}{\mu}\frac{\partial E_z}{\partial y}$$
$$\frac{\partial H_y}{\partial t} = \frac{1}{\mu}\frac{\partial E_z}{\partial x}$$
$$\frac{\partial E_z}{\partial t} = \frac{1}{\varepsilon}\left(\frac{\partial H_y}{\partial x} - \frac{\partial H_x}{\partial y} - \sigma E_z - J_z\right)$$

(1)

where, $H$ is the magnetic field strength, $E$ is the electric field strength, $\varepsilon$ is the dielectric permittivity (F/m), $\sigma$ is the conductivity (S/m), $\mu$ is the magnetic permeability (H/m), $t$ is the time (s), and $J_z$ is the source.

In this paper, we focus only on dielectric permittivity $\varepsilon$ and ignore the influence of conductivity $\sigma$ and magnetic permeability $\mu$ on the reflection coefficient. When the initial model $p_0(\varepsilon)$ is near the true model $p(\varepsilon)$, we split the true model $p(\varepsilon)$ into background model parameter $p_0(\varepsilon)$ and perturbed model parameter $\Delta p(\varepsilon)$, which can be expressed as $p(\varepsilon) = p_0(\varepsilon) + \Delta p(\varepsilon)$. Similarly, the wavefields $U(x,t,x_s)$ can also be split into background wavefields $U_0(x,t,x_s)$ and perturbed wavefields $\Delta U(x,t,x_s)$, which can be expressed as $U(x,t,x_s) = U_0(x,t,x_s) + \Delta U(x,t,x_s)$. When Taylor expands Equation (1) at $p = p_0$, using the Born approximation and ignoring higher order terms, the expression can be obtained as follows:

$$\mu\frac{\partial \Delta H_x}{\partial t} + \frac{\partial \Delta E_z}{\partial y} = -\Delta\mu\frac{\partial H_x}{\partial t}$$
$$\mu\frac{\partial \Delta H_y}{\partial t} - \frac{\partial \Delta E_z}{\partial x} = -\Delta\mu\frac{\partial H_y}{\partial t}$$
$$\varepsilon\frac{\partial \Delta E_z}{\partial t} - \frac{\partial \Delta H_y}{\partial x} + \frac{\partial \Delta H_x}{\partial y} + \sigma\Delta E_z = -\left(\Delta\varepsilon\frac{\partial E_z}{\partial t} + \Delta\sigma E_z\right)$$

(2)

When solving perturbed wavefields, information on background wavefields is generally required, so it is necessary to perform two forward simulations, which is also known as the reverse migration process. During the forward process of GPR simulation, the observed data $d$ and the relation of model parameter **m**, where **m** represents $\Delta$p, can be expressed as:

$$L\mathbf{m} = d$$

(3)

where, **m** is referred to as $\Delta\varepsilon$ when only the dielectric permittivity is considered; $L$ is the forward operator. We assume that $d_{obs}$ is the measured data, and $d$ is the perturbed wavefield. In LSRTM, the reflection coefficient is iteratively updated by inversion, and the objective function is established as follow [29]:

$$\min_{\mathbf{m}} S(\mathbf{m}) = \frac{1}{2}\|L\mathbf{m} - d_{obs}\|_2^2$$

(4)

where the simulated reflected wavefields must be satisfied with the Taylor expansion of Equation (1). The Lagrange multiplier method is utilized to solve the constrained optimization [30], and we have:

$$\min_{\mathbf{m}} J(\mathbf{m}) = \min_{\mathbf{m}} S(\mathbf{m}) + \int_{(x,y)\in H}\int_0^T \left(\psi_x e_1 + \psi_y e_2 + \phi e_3\right)dtdxdy$$

(5)

where $[\psi_x, \psi_y, \phi]$ are the Lagrange multiplier functions with the following form, where $e_1$, $e_2$, and $e_3$ are defined as:

$$e_1 = \mu \frac{\partial \Delta H_x}{\partial t} + \frac{\partial \Delta E_z}{\partial y} + \Delta \mu \frac{\partial H_x}{\partial t}$$

$$e_2 = \mu \frac{\partial \Delta H_y}{\partial t} - \frac{\partial \Delta E_z}{\partial x} + \Delta \mu \frac{\partial H_y}{\partial t} \qquad (6)$$

$$e_3 = \varepsilon \frac{\partial \Delta E_z}{\partial t} - \frac{\partial \Delta H_y}{\partial x} + \frac{\partial \Delta H_x}{\partial y} + \sigma \Delta E_z + \Delta \varepsilon \frac{\partial E_z}{\partial t} + \Delta \sigma E_z$$

Integrating Equation (5) by subsection integral method and combining initial conditions, termination conditions, and free boundary conditions, the adjoint equation can be expressed as:

$$\mu \frac{\partial \psi_x}{\partial t} = -\frac{\partial \phi}{\partial y}$$

$$\mu \frac{\partial \psi_y}{\partial t} = \frac{\partial \phi}{\partial x} \qquad (7)$$

$$\varepsilon \frac{\partial \phi}{\partial t} = \frac{\partial \psi_y}{\partial x} + \frac{\partial \psi_x}{\partial y} + \sigma \phi + \mathbf{m} \frac{\partial \phi}{\partial t} + (d_{obs} - d)$$

When only the dielectric permittivity $\varepsilon$ is considered, the gradients with respect to model perturbations can be expressed as Equation (8) and the termination conditions are expressed as Equation (9):

$$\frac{\partial J}{\partial \mathbf{m}} = \int_0^T \phi \frac{\partial E_z}{\partial t} dt \qquad (8)$$

$$[\psi_x, \psi_y, \phi]_{t=T} = 0, \ \frac{\partial [\psi_x, \psi_y, \phi]_{t=T}}{\partial t} = 0 \qquad (9)$$

### 2.2. MTV Normalization

In the traditional TV method, the sensitivity to smoothing parameters is significantly high due to nonlinearity caused by data fitting, regularization terms, and convergence. As a result, the LSRTM problem becomes highly unstable, and its convergence cannot be guaranteed. The objective function of the LSRTM-TV is as follows:

$$S^e(\mathbf{m}) = \min_{\mathbf{m}} \left\{ \|d_{obs} - L\mathbf{m}\|_2^2 + \lambda \|\mathbf{m}\|_{TV} \right\} \qquad (10)$$

where $\| \cdot \|_{TV}$ is the TV regularization operator as follows:

$$\|\mathbf{m}\|_{TV} = TV(\mathbf{m}) = \int_\Omega |\nabla \mathbf{m}| d\Omega \qquad (11)$$

where $\Omega$ is the imaging region, $|\cdot|$ is the absolute value sign, and $\nabla$ is the gradient operator. Since the derivative of the TV operator is non-continuous, it is guaranteed to be differentiable by the following approximation:

$$TV_\delta(\mathbf{m}) = \int_\Omega \sqrt{|\nabla \mathbf{m}|^2 + \delta^2} d\Omega \qquad (12)$$

Hence, we introduce a new LSRTM method with MTV regularization to improve the imaging resolution and mitigate artifacts. The objective function of the LSRTM-MTV method is given as follows:

$$S^e(\mathbf{m}, \mathbf{u}) = \min_{\mathbf{m}, \mathbf{u}} \left\{ \|d_{obs} - L\mathbf{m}\|_2^2 + \lambda_1 \|\mathbf{m} - \mathbf{u}\|_2^2 + \lambda_2 \|\mathbf{u}\|_{TV} \right\} \qquad (13)$$

where, $\lambda_1$ and $\lambda_2$ are both positive regularization parameters. Compared with a conventional *TV* regularization term, Equation (13) contains a new variable $\mathbf{u}$ and an additional regularization term. The objective function can be written as:

$$S^e(\mathbf{m}, \mathbf{u}) = \min_{\mathbf{u}} \left\{ \min_{\mathbf{m}} \left\{ \|d_{obs} - L\mathbf{m}\|_2^2 + \lambda_1 \|\mathbf{m} - \mathbf{u}\|_2^2 \right\} + \lambda_2 \|\mathbf{u}\|_{TV} \right\} \qquad (14)$$

As can be seen from Equation (14), the regularization parameter $\lambda_1$ controls the trade-off between the data misfit term and the Tikhonov regularization term, and $\lambda_2$ is employed here to balance the amount of interface preservation in LSRTM. The alternating-minimization algorithm is used to solve the dual minimization problem [31]. By using an initializer $\mathbf{u}^{(0)} = \mathbf{m}^{(0)}$ to solve Equation (14), the solutions of two minimization sub-problems can be obtained:

$$\mathbf{m}^{(k)} = \underset{\mathbf{m}}{\mathrm{argmin}}\{S_1^e(\mathbf{m})\} = \underset{\mathbf{m}}{\mathrm{argmin}}\left\{\|d_{obs} - L\mathbf{m}\|_2^2 + \lambda_1\|\mathbf{m} - \mathbf{u}^{(k-1)}\|_2^2\right\} \tag{15}$$

$$\mathbf{u}^{(k)} = \underset{\mathbf{m}}{\mathrm{argmin}}\{S_2^e(\mathbf{m})\} = \underset{\mathbf{u}}{\mathrm{argmin}}\left\{\|\mathbf{m}^{(k)} - \mathbf{u}\|_2^2 + \lambda_2\|\mathbf{u}\|_{TV}\right\} \tag{16}$$

where $k$ is the iteration step. Noting that the two sub-problems correspond to different parts, the first is to solve for $\mathbf{m}^{(k)}$ by using LSRTM with the Tikhonov regularization [32] and prior image $\mathbf{u}^{(k-1)}$; while the second is to solve for $\mathbf{u}^{(k)}$ using the $L^2$-TV minimization to preserve the outline sharpness of interfaces within the LSRTM image $\mathbf{m}^{(k)}$, as well, it also denoises the image. In this paper, the Limited-memory Broyden–Fletcher–Goldfarb–Shanno (L-BFGS) algorithm [33] is employed to solve for m(k) in the sub-problems (15), while the Split–Bregman method [34] is utilized to solve for u(k) in the sub-problems (16).

The iterative formula for **m** is as follows:

$$\mathbf{m}^{(k+1)} = \mathbf{m}^{(k)} + \alpha^{(k)}\mathbf{p}^{(k)} \tag{17}$$

where $\mathbf{p}^{(k)}$ is the updating direction and $\alpha^{(k)}$ is the iteration step size. Initially, the L-BFGS method is employed to obtain parameter $\mathbf{p}^{(k)}$ using the calculated function $\mathbf{g}$ based on Equation (8). In this paper, we employ an imprecise line search method based on the strong Wolfe criteria, which requires satisfying the following two conditions:

$$\phi(\alpha) \leq \phi(0) + c_1\alpha\phi'(0) \tag{18}$$

$$|\phi'(\alpha)| \leq c_2|\phi'(0)| \tag{19}$$

where $\phi(\alpha) = S(\mathbf{m}_k + \alpha\mathbf{p}_k)$, $0 < c_1 < c_2 < 1$, $c_1 = 10^{-4}$, $c_2 = 0.9$.

For the L-BFGS algorithm, it is recommended to initially try step size $\alpha^{(0)} = 1$ in the line search algorithm, as it can provide the L-BFGS algorithm with a superlinear convergence rate. During the initial iterations, the scaling in the descent direction may be small due to the potential inadequacy of the Hessian approximation. Thus, the step size $\alpha^{(k0)}$ is calculated using the following formula:

$$\alpha^{(k0)} = \frac{S_k}{\gamma\mathbf{g}} \tag{20}$$

where $\gamma$ is a regulatory factor which is related to the antenna frequency.

The line search algorithm may appear computationally complex and time-consuming. However, the computational cost is contingent upon the availability of a suitable $\mathbf{p}^{(k)}$ in the descent direction, and $\phi(\alpha)$ can be approximated by a quadratic or cubic function. The model **m** is then updated in Equation (17).

The proposed LSRTM-MTV method has two major advantages. First, the original optimization problem in LSRTM is decoupled into two simple subproblems; solving the two sub-problems in Equations (15) and (16) is much simpler than solving the original LSRTM problem. Second, solving for $\mathbf{u}^{(k)}$ is a $L^2$-TV denoising problem. Many robust computational methods are available for solving this minimization problem; therefore, the LSRTM-MTV method is robust and has a fast convergence rate.

The LSRTM-MTV workflow is outlined in Algorithm 1 and depicted in Figure 1. It can be observed that the workflow of LSRTM-MTV is similar to full waveform inversion [35]. The main difference is that LSRTM-MTV takes reflected wave information as input and

output reflection coefficients, while full waveform inversion takes the full-wavefield information as input and output model parameters.

---

**Algorithm 1**: LSRTM-MTV Algorithm

---

    Where Equation (2) is satisfied
Calculate Equation (1) for ΔU; Calculate Equation (7) for Ez and ϕ;
    if Equation (9) is satisfied
      return Ez and ϕ;
    end
Calculate Equation (8) for g;
    Use Wolfe conditions to get the iteration step size;
    Use L-BFGS to update the reflection coefficient.
    Load MTV and denoise the results with Equations (14)–(16);
    end
    Stop until the convergence condition is satisfied.

---

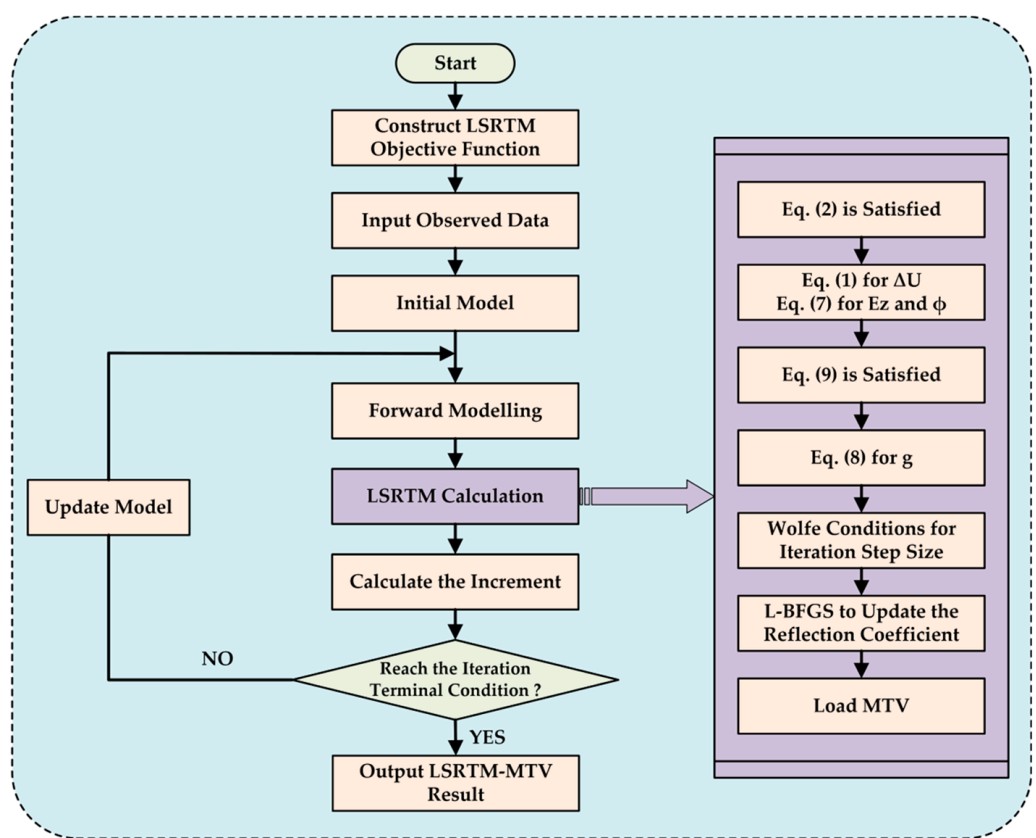

**Figure 1.** Flowchart of the proposed LSRTM-MTV algorithm.

## 3. Numerical Examples

A model is established to verify the applicability of the LSRTM-MTV method to complex models [36], the size of which is 4.0 m × 8.0 m, as shown in Figure 2a. Two-layer undulating interfaces are designed in the model, with two lithologic contrast units on the right and two irregular cavities in the lower layer. The simulation is performed under the common-offset mode of the monostatic system, and the transmitting and receiving antennas are placed in the air layer. The smoothed true model is used as the initial model for migration. To obtain the initial model, a Gaussian filter with a template size of 50 × 50 and a standard deviation of 3 was used for smoothing; note that these two parameters determine the degree of smoothness compared to the true model. The disturbance model, calculated by the true model and the smoothed model, is shown in Figure 2b.

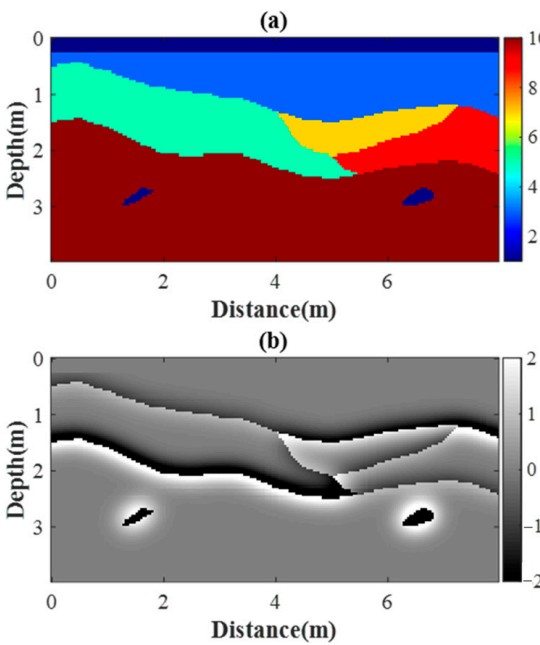

**Figure 2.** True model (**a**) and the corresponding disturbance model (**b**).

The simulation was carried out using the finite difference time domain (FDTD) method [2,37] with a Riker wavelet source at the central frequency of 200 MHz and a time window of 108 ns. The time and space intervals were set at 0.09 ns and 0.04 m, respectively. The simulation results of the true model forward record and the initial model forward record are shown in Figure 3a,b, respectively. The residual record of the true model and initial model are shown in Figure 3c, confirming the feasibility and appropriateness of the smoothed model as an initial model.

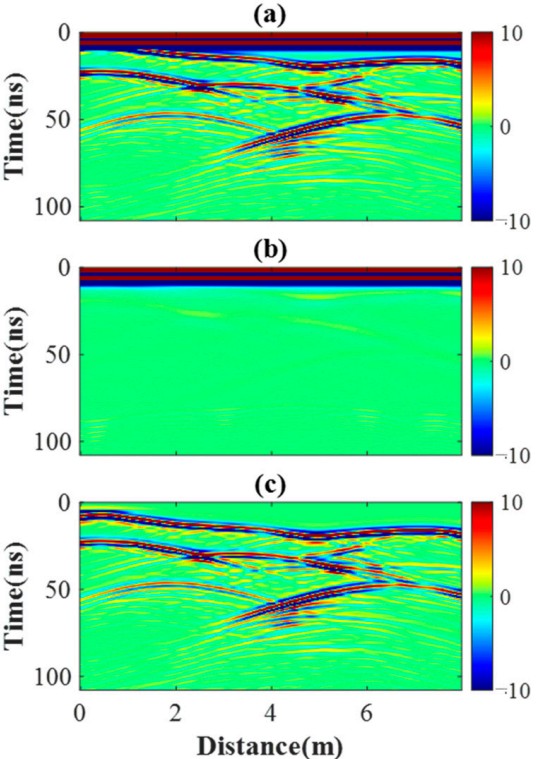

**Figure 3.** True model forward record (**a**), initial model forward record (**b**), and residual record (**c**).

The imaging results are shown in Figure 4. LSRTM imaging results with 20, 40 and finial iterations are shown in Figure 4a–c, respectively. It can be seen that imaging results after the 20th iteration are relatively vague, as the interface and abnormal position cannot be precisely localized, while with increasing iterations the interface and abnormal position can be pre-determined after the 40th iteration. As shown in the finial iteration, except for significant noise and artifacts, the image result clearly shows the fluctuating interface, the interface of lithological units, and the location of two irregular anomalies.

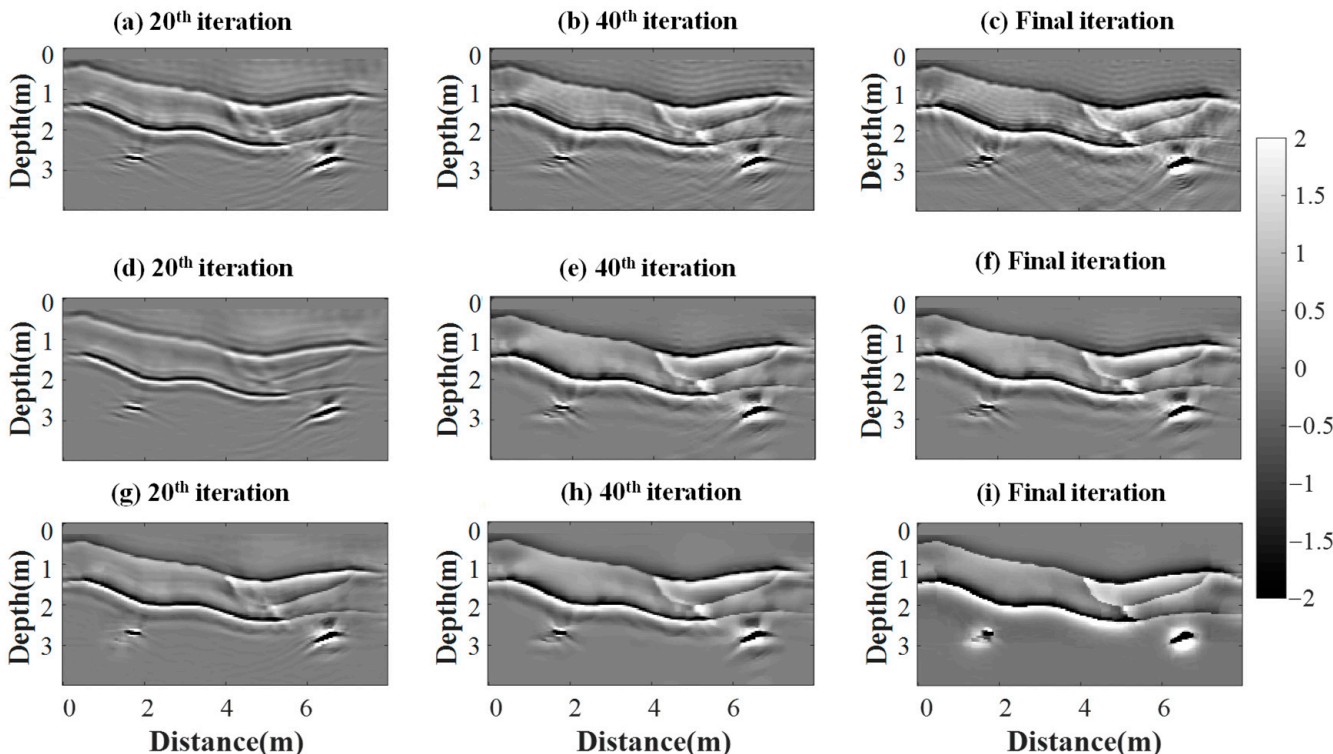

**Figure 4.** LSRTM imaging results (**a**–**c**), LSRTM-TV imaging results (**d**–**f**) and LSRTM-MTV imaging results (**g**–**i**) with different iterations.

To this end, MTV regularization is introduced as an optimization scheme, and the conventional TV is also presented to compare performance after processing the iterative results of Figure 4a–c. Imaging results of LSRTM-TV with 20, 40 and finial iterations are shown in Figure 4d–f and LSRTM-MTV with 20, 40 and finial iterations are shown in Figure 4g–i, respectively. Compared to LSRTM results, the imaging results of traditional TV show some noise suppression capabilities while artifacts remain, blurring the interfaces. In contrast, MTV imaging results demonstrate enhanced noise suppression performance. For example, see the comparative imaging results of the 20th iteration shown in Figure 4f,i; the imaging resolution is significantly improved as the reflection interface is clearer, and the abnormal position is localized precisely, allowing verification of the effectiveness of the LSRTM-MTV scheme.

The convergence curve using different methods is shown in Figure 5. As a result of the highly unstable convergence encountered in the LSRTM-TV problem, the convergence of the calculation process is not assured, leading to termination after a mere 42 iterations. In contrast, the MTV method proposed in this paper addresses the limitations of the TV method.

To assess the quality of the imaging results, the mean square error (MSE) was calculated between each of the three results and the disturbance model. MSE is the most common estimator of image quality measurement metrics. It is a full reference metric, and values closer to zero are better [38]. MSE between two images is defined as:

$$MSE(x,y) = \frac{1}{N}\sum_{i=1}^{N}(x_i - y_i)^2 \tag{21}$$

where *x* represents the disturbance model, *y* represents the imaging results of the three methods, and *N* represents the number of data matrix elements.

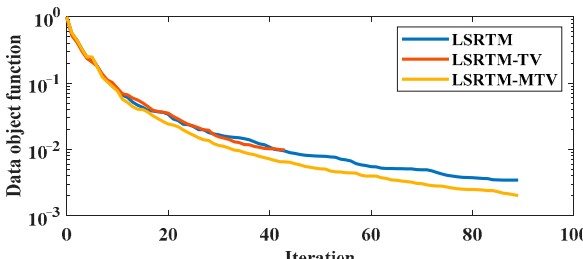

**Figure 5.** Convergence curve with LSRTM, LSRTM-TV, and LSRTM-MTV.

Compared to the LSRTM method, LSRTM-MTV demonstrates a 30.4% enhancement in computation speed and an 21.1% reduction in MSE, as shown in Table 1. Additionally, the amplitude curve of track 85 was randomly selected and plotted in Figure 6. The amplitude curve in Figure 6 shows the disturbance model as a solid black line, while the LSRTM, LSRTM-TV, and LSRTM-MTV results are shown as solid green, dashed blue, and dashed red lines, respectively. Although the LSRTM results generally approximate the reflection interface positions in the disturbance model, they exhibit significant noise and artifacts in the uniform medium. After applying TV processing, the artifacts are somewhat reduced, and the oscillations are eliminated. After MTV processing, the red dashed line significantly reduces the noise and artifacts without sacrificing the interface information. Overall, the results are in good agreement with the disturbance model.

**Table 1.** Comparison of calculation parameters between three methods.

|  | LSRTM | LSRTM-TV | LSRTM-MTV |
|---|---|---|---|
| MSE | 0.0649 | 0.0593 | 0.0512 |
| Iterations | 90 | 42 | 90 |
| Time (min) | 33.9255 | 11.9032 | 23.5991 |

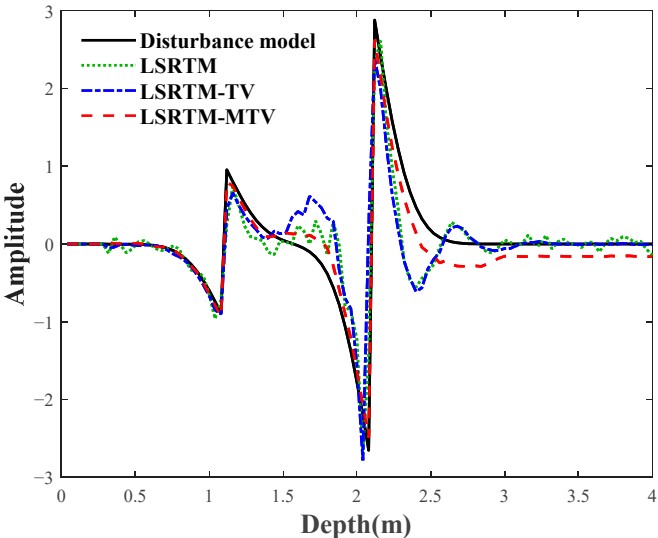

**Figure 6.** Multiple curve comparison with disturbance model, LSRTM, LSRTM-TV, and LSRTM-MTV.

To determine the effect of $\lambda_1$ on the imaging results, we set $\lambda_1$ as 0.0001, 0.001, and 0.01 to carry out comparative experiments, while $\lambda_1$ values of 0.1, 10, and 100 resulted in unstable and inaccurate convergence.

Figure 7 displays the imaging results for different $\lambda_1$ values. It can be observed that the selection of $\lambda_1$ significantly influences the imaging effect of LSRTM-MTV. Clutter is removed to the maximum extent and the interface information is retained when $\lambda_1$ is set to 0.01, achieving the best effect. The outcomes presented in Table 2 further indicate that the lowest MSE is achieved when $\lambda_1 = 0.01$. Subsequent experiments were developed based on this result, using $\lambda_1 = 0.01$. It should be noted that $\lambda_2$ in Equation (14) is determined by the data itself and is not the focus of our discussion.

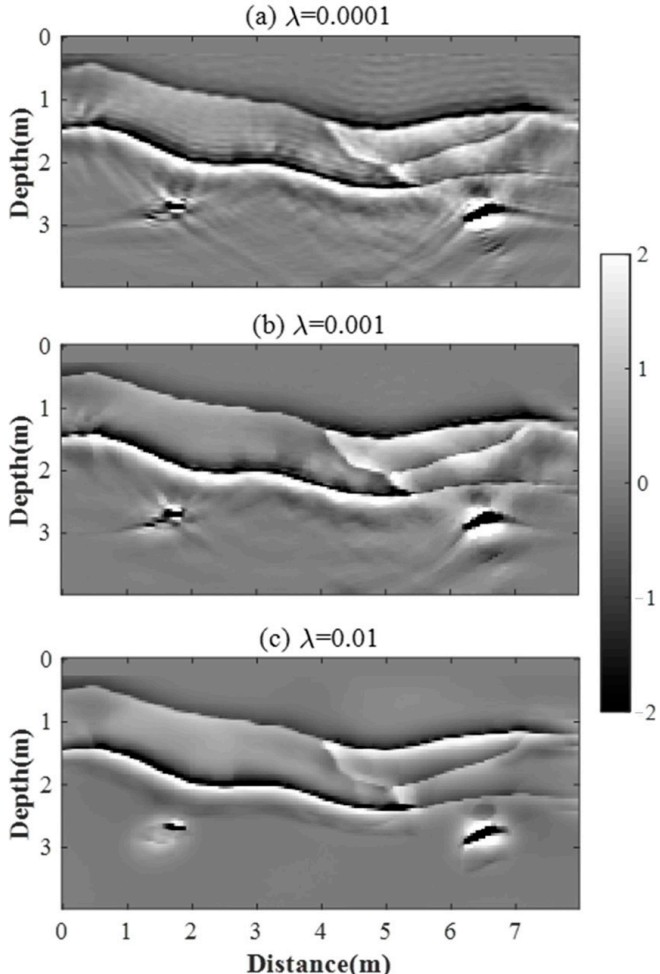

**Figure 7.** Different $\lambda$ imaging results with (**a**) $\lambda = 0.0001$, (**b**) $\lambda = 0.001$, (**c**) $\lambda = 0.01$.

**Table 2.** Comparison of MSE between different $\lambda_1$.

|  | $\lambda_1 = 0.0001$ | $\lambda_1 = 0.001$ | $\lambda_1 = 0.01$ |
|---|---|---|---|
| MSE | 0.0641 | 0.0563 | 0.0512 |

To evaluate the robustness of the proposed method, we added 10 dBW Gaussian white noise to the forward recording and tested its anti-noise performance. The forward recording after removing the direct wave is shown in Figure 8. As shown in Figure 9 and Table 3, the imaging result obtained using LSRTM-MTV is superior to that obtained using traditional TV, even after adding Gaussian white noise. This result further verifies the robustness and anti-noise performance of the proposed method.

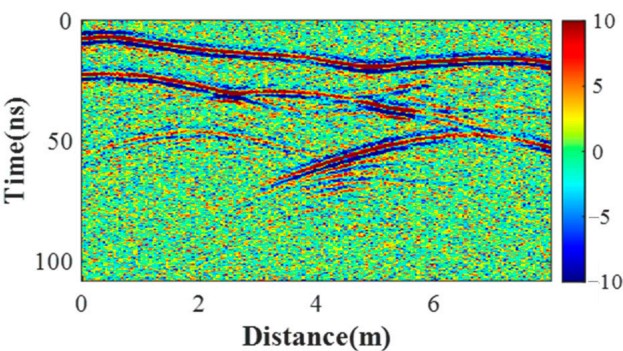

**Figure 8.** Forward recording after noise (residual record).

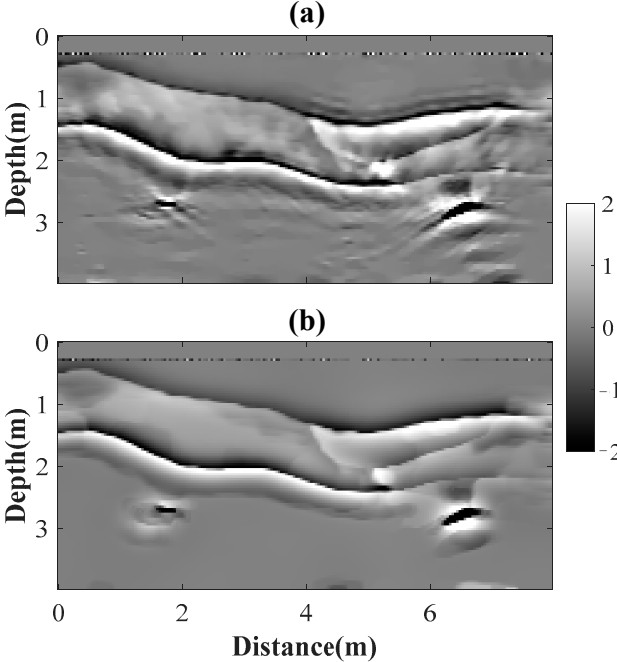

**Figure 9.** (**a**) LSRTM-TV imaging result of noised record; (**b**) LSRTM-MTV imaging result of noised record.

**Table 3.** Comparison of MSE with noised record between LSRTM-TV and LSRTM-MTV.

|  | LSRTM-TV | LSRTM-MTV |
|---|---|---|
| MSE | 0.0667 | 0.0572 |

## 4. Laboratory Data Experiment

To demonstrate the applicability of the proposed method, a laboratory experiment was conducted using a sand tank model at Guilin University of Technology, as shown in Figure 10. The sandpit contained five abnormal bodies including two empty pipes on the left, cuboid empty abnormal bodies in the center, and two solid pipes on the right. The sandpit was divided into three levels, with dielectric constants of 3.0 and 2.4 for the first and second levels, respectively, while the bottom level represented the sandpit boundary. The schematic diagram of the laboratory model is shown in Figure 11. The parameters of abnormal are listed in Table 4. For data acquisition, we used a 900 MHz antenna of a GSSI SIR-4000 radar in point measurement mode; the sampling points per trace and the time window were set as 512 and 10 ns, respectively. The channel spacing was set to 0.004 m, and the length of the survey line was 2.5 m, and a total of 625 traces were collected, as shown by the B-scan in Figure 12.

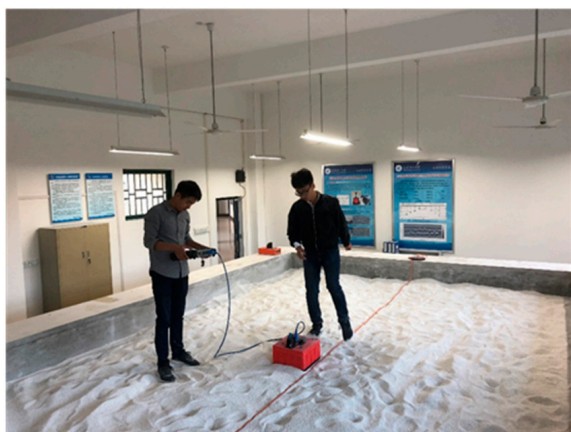

**Figure 10.** Physical laboratory model.

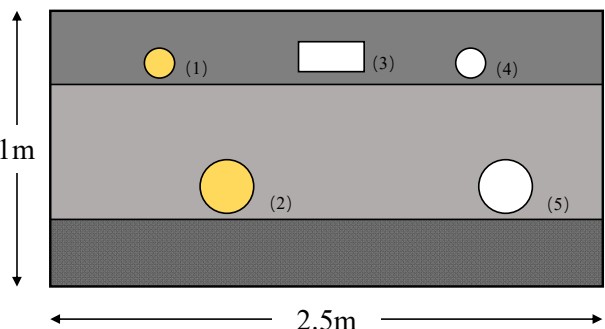

**Figure 11.** The schematic diagram of laboratory model.

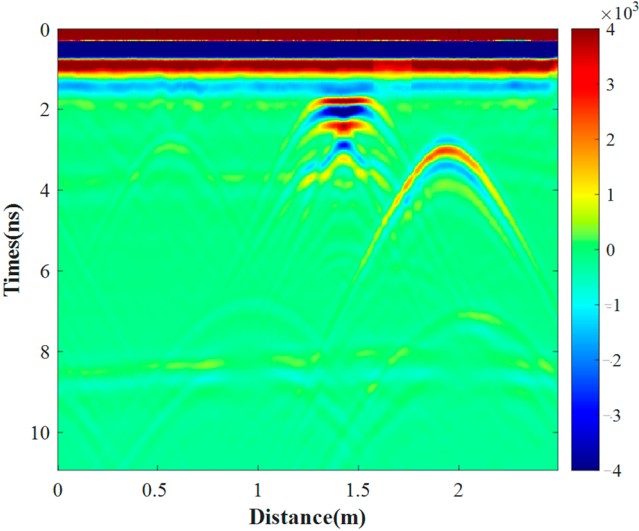

**Figure 12.** The original B-scan profile of laboratory data.

**Table 4.** The dielectric constants of the laboratory model.

| Number | Material | Size (cm) | Burial Depth (cm) | Dielectric Constant |
|---|---|---|---|---|
| 1 | Pipes (solid) | 5 (radius) | 22 | 4.0 |
| 2 | Pipes (solid) | 10 (radius) | 50 | 4.0 |
| 3 | Cuboid abnormal (empty) | 20 × 10 | 15 | 1.0 |
| 4 | Pipes (empty) | 5 (radius) | 20 | 1.0 |
| 5 | Pipes (empty) | 10 (radius) | 50 | 1.0 |

In the original B-scan profile, the reflected intensity was conjointly determined by two factors: the burial depth and the difference in relative permittivity between the anomalies and the background medium. Essentially, as the burial depth increases, the attenuation of electromagnetic wave energy becomes more pronounced, leading to a decrease in the energy of the reflected wave. Simultaneously, a higher contrast in relative permittivity between the anomalies and the background leads to a greater disparity in wave impedance, consequently amplifying the reflected energy.

As observed in Figure 12, various nuances become apparent when considering these factors. Anomaly 1, despite its relatively shallow burial depth, had a minimal wave impedance difference, resulting in weaker reflected wave energy. Anomaly 2, buried deeper and with a smaller wave impedance difference, resulted in the weakest reflected wave energy. In stark contrast, anomalies 3 and 4, with shallower burial depths and greater wave impedance differences, exhibited the strongest energy in the reflected waves. Anomaly 5, although buried deeper, outperformed anomaly 2 in reflected wave energy owing to a more pronounced wave impedance difference.

To accurately locate and depict the subsurface anomalies, the proposed LSRTM-MTV method, introduced in this paper, was utilized to process the collected data. Considering the stratification of the sandpits, a two-layer uniform initial model was established to optimize the imaging results and compare them with conventional LSRTM method, as shown in Figure 13. While both methods were able to generally identify the presence of five subsurface anomalies, the conventional LSRTM method exhibited significant background clutter. This clutter complicated the precise localization and morphological characterization of these anomalies, as shown in Figure 13a. In contrast, LSRTM-MTV remarkably mitigated such background clutter, leading to a far more accurate representation of the anomalies. Specifically, Figure 13b illustrates that the enhanced method allowed for a clearer delineation of the upper and lower interfaces of anomalies 2 and 3. This improvement is highlighted by white arrows pointing to the lower interfaces of these anomalies.

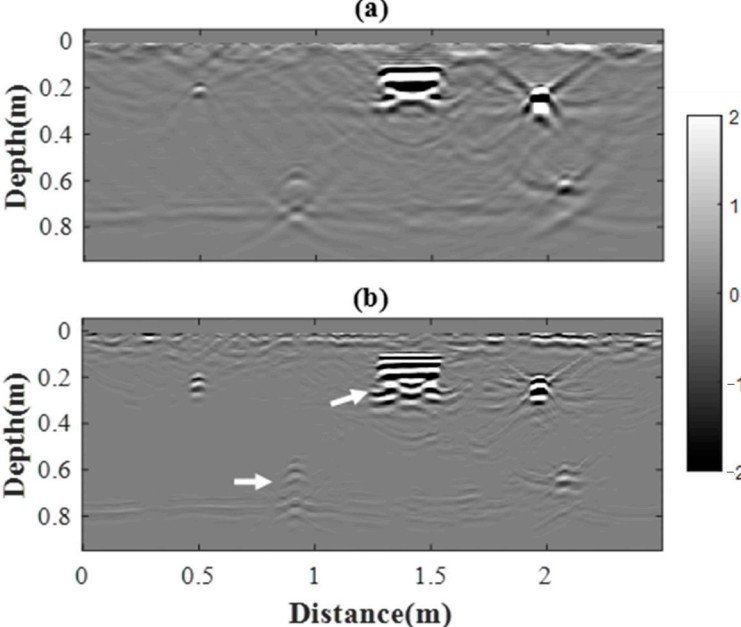

**Figure 13.** LSRTM imaging results: (**a**) LSRTM-MTV imaging result; (**b**) laboratory data.

The results thus conclusively demonstrate that the proposed LSRTM-MTV method significantly outperforms the conventional LSRTM approach when applied to laboratory data. These findings substantiate the efficacy and robustness of the LSRTM-MTV method, highlighting its utility for accurate subsurface anomaly detection in practical applications.

## 5. Discussion

This paper introduces the use of MTV methods to enhance the traditional LSRTM, with the objective of improving the imaging quality of subsurface structures. In comparison to the conventional TV method, MTV offers notable advancements in stability and accuracy while maintaining a computationally efficient framework.

However, it is important to acknowledge the limitations inherent in the current study. Specifically, the work was confined to permittivity contrast media models, assumed uniform background media, and was highly sensitive to the selection of the parameter $\lambda_1$, which substantially influences imaging efficacy. Future work should address these limitations by exploring models that incorporate dispersive or anisotropic media, thus aligning more closely with real-world scenarios. In addition, rigorous experimentation on parameter selection is essential to validate the robustness of the proposed method.

Moreover, the LSRTM-MTV algorithm demonstrates potential applicability beyond the scope of this study, particularly in other imaging techniques requiring the minimization of similar objective functions. Future studies could further investigate these avenues, possibly engaging in comparative assessments with other state-of-the-art techniques to establish the method's relative merits.

## 6. Conclusions

In this study, the Born approximation method was used to derive the LSRTM process first. Next, the gradient of the true model and the perturbation model was calculated utilizing a method similar to the RTM, and the Lagrange multiplier method was used to update the reflection coefficient model. Finally, the methodology was integrated into the data processing of the GPR application. To further improve the imaging resolution, we then introduced MTV regularization into the LSRTM process as an optimization scheme to reduce artifacts.

The numerical experiment results demonstrated that the imaging result of LSRTM-MTV outperforms the LSRTM in terms of noise suppression and artifact elimination. Via LSRTM-MTV processing, the undulation interface and the location of the irregular anomalies were sufficiently characterized with a high convergence rate. Compared with LSRTM, the computation speed of LSRTM-MTV is higher by 30.4%, and the MSE is reduced by 21.1%. This implies that the proposed LSRTM-MTV has considerable potential for GPR applications for refined delineation of the undulation interface and irregular anomalies. The laboratory data experimental results demonstrated the efficacy and robustness of the proposed method in practical applications.

**Author Contributions:** Conceptualization, Q.D. and S.W.; methodology, S.W. and Y.L.; software, S.W. and Y.L.; formal analysis, Q.D., S.W. and Y.L.; investigation, Y.L.; data curation, S.W. and Y.L.; writing—original draft preparation, Y.L.; writing—review and editing, S.W. and Y.L.; supervision, Q.D.; funding acquisition, Q.D. and Y.L. All authors have read and agreed to the published version of the manuscript.

**Funding:** This research was supported by the National Natural Science Foundation of China (grant Nos. 41874148 and 42174178), National Key Research and Development Program of China (grant Nos. 2018YFC0603903), and in part by the Postdoctoral Science Foundation of Central South University under grant 22021133.

**Institutional Review Board Statement:** Not applicable.

**Informed Consent Statement:** Not applicable.

**Data Availability Statement:** Data sharing is not applicable to this article.

**Acknowledgments:** We thank Hai Liu from Guangzhou University for the field data.

**Conflicts of Interest:** The authors declare no conflict of interest.

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
