# Peer review of "Least Squares Reverse Time Migration of Ground Penetrating Radar Data Based on Modified Total Variation"

_applsci, doi:10.3390/app131810028_

Round 1
Reviewer 1 Report
The manuscript details a laboratory experiment that utilizes the LSRTM-MTV method for processing Ground Penetrating Radar (GPR) data to detect subsurface anomalies. This experiment was conducted using a sand tank model at Guilin University of Technology. The methodology is soundly based on the Born approximation method, with efforts to enhance imaging resolution through MTV regularization.
Strengths:
1. Methodology: The approach, integrating the LSRTM process with MTV regularization to enhance GPR data imaging resolution, is innovative and holds practical significance.
2. Laboratory Set-up: The sandpit model is well-detailed with clearly defined dielectric constants and structures. This comprehensive description will allow other researchers to replicate the experiment or adapt it for their own studies.
3. Comparison with Conventional Methods: The comparison of results from the LSRTM-MTV method with those from the conventional LSRTM method provides a clear perspective on the improvements brought about by the former.
Areas for Improvement and Suggestions:
1. Clarity and Presentation: It may be beneficial to ensure that all figures (e.g., Figure 9, 10, 11) are clear and adequately labeled. This will assist readers in understanding the experimental setup and results.
2. Anomalies Analysis: The manuscript notes differences in reflected intensity between various anomalies. A more in-depth analysis or discussion on why anomalies 3 and 4 have stronger reflected intensity compared to anomalies 1 and 2 could provide more insight.
3. Performance Metrics: While qualitative observations about the superiority of the LSRTM-MTV method are mentioned, the inclusion of quantitative performance metrics, such as Signal-to-Noise Ratio (SNR) or a similar metric, might bolster the paper's claims.
4. Future Work: The conclusion touches upon potential future works like focusing on the permittivity contrast medium model. It would be beneficial if the authors could elaborate more on the challenges and possible methodologies they envision for these future studies.
5. Computational Efficiency: The paper mentions "faster computation speed" and "fewer iterations". It would be beneficial to provide some actual computational times or statistics to back up these claims.
Overall, the paper presents a methodological advancement in the processing of GPR data using the LSRTM-MTV method. While the results are promising and the experiment setup is detailed, there are areas where additional information, analysis, or clarification could enhance the manuscript's impact. By addressing these suggestions, the authors can present a more comprehensive and convincing case for the efficacy of their proposed method.
The manuscript is generally well-written. However, there are areas where the language can be improved for clarity, precision, and style: use of articles, active/passive.
Author Response
Dear Editor and Reviewer:
We would like to sincerely thank the Reviewer and the Associate Editor for their time and effort spent reviewing our work. Your comments were very fruitful and helped us improve the manuscript. The revisions in the manuscript are highlighted using red color. We hope our responses have now clarified any pending issue. For your guidance, itemized response to Reviewer’s comments has been made point by point. Please see the attachment.

Reviewer 2 Report
Dear Authors
I comment you on a good work. The only minor revision required is:
Abstract and Conclusion should be supported by numbers. The general terms like better or outperform are not enough when comparing models.
Author Response

(The authors gave the same response as above.)

Reviewer 3 Report
The paper presents an innovative approach to improve the imaging resolution of GPR data using LSRTM with MTV regularization. The numerical examples and experimental results provided effectively validate the proposed method's efficacy. However, some sections could be further refined for clarity and completeness.
The paper could benefit from discussing potential future work or extensions of the proposed method, including addressing limitations, exploring real-world data, and comparing with other state-of-the-art techniques.
The mention of a laboratory data experiment towards the end of the abstract is intriguing; however, the section is incomplete and lacks detailed explanation. It would be beneficial to expand upon this experiment and provide more insights into its results and implications.
I would find it satisfactory if accompanied by a well-constructed block diagram illustrating the proposed approach, which should also be thoroughly elucidated.
Quality of english is satisfactory. I would request 2-3 round of reviews by the authors to improve it further.
Author Response

(The authors gave the same response as above.)

Reviewer 4 Report
Please see the attached report.

Minor editing of English language required.
Author Response

(The authors gave the same response as above.)

Reviewer 5 Report
Dear Editor,
The reviewer reviewed the manuscript (MS) titled “Least Squares Reverse Time Migration of GPR Data Based on Modified Total Variation” submitted to Applied Sciences journal in detail to meet the scientific requirements.
In the MS, the authors proposed a new LSTRM based on MTV regularization coupling objective function and constraints. They validated their results using synthetic and laboratory data.
General Comments:
In title, full form of GPR should be given.
In Abstract, optimization methodology, results and performance of the study should be given. Motivation/future works or concluded remarks may be included.
In line 58, TV?
Literature and findings is not introduced properly in the Introduction section. 24 references are cited, but an efficient literature review is not provided.
Research question should be introduced, and motivation and novelty should be clearly explained. It is not sufficient to mention about research gap.
In Eq. 3, Dp does not exist, as mentioned in line 103. m is Dp should be mentioned before Eq. 3.
e1, e2, e3 in Eq. 6 should be introduced.
Full form of BFGS should be given where the reader first meets.
Objective functions are presented, but constraints are not clearly defined. The readers of the journal may need to understand regularizations.
Which optimization schemes are adopted? Clearly define. Are the results local or global?
The authors presented errors, iteration numbers, etc. Sensitivity of the objective function should be given. Because this issue is important to avoid equifinality problem.
Discussion section is required. Findings of this study and the other research should be compared. Advantages/disadvantages, shortfalls should be discussed.
In line 346, letter?
Conclusion section is a summarization of the MS itself. Conclusion remarks, future works, etc. should be highlighted in this section.
Author Response

(The authors gave the same response as above.)

Round 2
Reviewer 1 Report
The authors improved the manuscript as suggested.
Reviewer 4 Report
This paper may be accepted in its present form.
Minor editing of English language required
Reviewer 5 Report
This manuscript can now be accepted for publication.